# Evaluation of Subcortical Structure Volumes in Patients with Non-Specific Digestive Diseases

**DOI:** 10.3390/diagnostics12092199

**Published:** 2022-09-09

**Authors:** Katarzyna Skrobisz, Grazyna Piotrowicz, Agata Rudnik, Patrycja Naumczyk, Agnieszka Sabisz, Karolina Markiet, Edyta Szurowska

**Affiliations:** 1Department of Radiology, Medical University of Gdansk, 80-214 Gdansk, Poland; 2Department of Gastroenterology, Self-Dependent Health Care Unit of Ministry of Interior, 80-104 Gdansk, Poland; 3Department of Clinical Psychology and Health, Institute of Psychology, University of Gdansk, 80-309 Gdansk, Poland; 4Academic Psychological Support Center, University of Gdansk, 80-309 Gdansk, Poland; 52nd Department of Radiology, Medical University of Gdansk, 80-214 Gdansk, Poland

**Keywords:** functional dyspepsia (FD), irritable bowel syndrome (IBS), colitis ulcerosa, Crohn’s disease, brain-gut axis (BGA)

## Abstract

(1) Background: To evaluate volume of subcortical structures such as hippocampus, globus pallidus, putamen, thalamus, nucleus accumbens, amygdala, caudate in patients with non-specific digestive diseases (functional dyspepsia—FD, irritable bowel syndrome—IBS) and non-specific inflammatory bowel diseases—IBD (colitis ulcerosa and Crohn’s disease) in comparison to healthy control group (CON). (2) Material: The analysis included data obtained from 57 patients (FD-18, IBS-20, IBD-19) and 19 persons in control group. Both groups underwent examination in a 3T scanner (Achieva TX Philips Healthcare). (3) Results: Significant differences between the IBD group and Control group in volume of left thalamus and IBD group vs Control group in volume of right thalamus. (4) Conclusions: The brain-gut axis hypothesis explains connection between biological behavior, emotions and cognitive functions in patients with gastrointestinal disease. We found that there is a difference between volume of thalamus in IBD patients in comparison to both IBS and control group and it occurred to be smaller. Excess inflammation can be linked with psychological disorders like depressive symptoms, sleep difficulties and/or fatigue. Therefore, there is a need for using treatment both for depressive symptoms and IBD to reduce the causes and effects of inflammation.

## 1. Introduction 

Both non-specific functional gastrointestinal disorders—FGIDs (including irritable bowel syndrome, IBS and functional dyspepsia, FD) and non-specific inflammatory bowel diseases—IBD (ulcerative colitis, UC and Crohn’s disease, CD) are chronic gastrointestinal (GI) disorders severely affecting the patient’s quality of life. The aetiology of FD, IBS and IBD is multifactorial (inc. genetic risk and environmental factors) and still unclear but it is also considered to be closely related to psychosocial factors like stress, depression, anxiety etc. [1,2,3,4].

Rome IV is a compendium of criteria covering the symptoms of irritable bowel syndrome and functional dyspepsia [5]. IBS is described as a chronic and disabling functional bowel disorder. Functional label states for the situation when no visible structural or biochemical abnormalities are found. The diagnosis of IBS is based on recurrent abdominal pain related to defecation or along with a change in stool frequency or form [6], while functional dyspepsia is characterized as relapsing and remitting disorder consisting of a sensation of pain or burning in the epigastrium, early satiety and fullness during or after a meal [7]. The term inflammatory bowel diseases covers two chronic, non-specific conditions: Crohn’s disease (CD) and ulcerative colitis (UC) with recurring periods of flare-ups and remissions. They cause progressive bowel damage and require lifelong treatment [8] and IBD patients experience distressing gastrointestinal symptoms like diarrhea, weight loss, abdominal pains or nausea or exclusion of some dietary products due to their subjective low tolerance [9]. Curiously, CD and UC patients in phases of clinical remission can possibly suffer IBS-like symptoms, implying that IBS and IBD perhaps share both common risk factors and alterations of the brain-gut axis (BGA) [4].

A bidirectional communication pathway between the gut and brain is maintained via a network consisted of the central nervous system (CNS), autonomic nervous system (ANS) and enteric nervous system (ENS) as well as hypothalamic–pituitary–adrenal (HPA) axis, neural, endocrine and immune systems [10,11]. The brain-gut axis hypothesis explains the connection between biological behavior, emotions and cognitive functions in patients with gastrointestinal diseases [12].

The stress response involves different regions within the brain in particular the amygdala, hippocampus and hypothalamus. Recent studies indicate that the effects of stress in IBD can be mediated among others through HPA axis function and alterations in bacterial-mucosal floral interactions [13,14,15]. The HPA axis stands for a major axis of the neuroendocrine system and controls our reactions to stress. Its dysregulation has been connected to mood disorders like depression, anxiety, or bipolar disorder [16]. The activated HPA axis causes the secretion of corticotropin-releasing factor (CRF) from the hypothalamus, which stimulates the pituitary gland to release adrenocorticotropic hormone (ACTH) which triggers the immunosuppressive stress-hormone cortisol from the adrenal cortex and that leads to the synthesis of anti-inflammatory cytokines [14,17]. Crypt analyses from rodents and humans proved that stress-induced cortisol increases intestinal barrier dysfunction, as well as the role for cortisol was shown in regulating intestinal inflammation and altering microbiota composition [18]. Mawdsley et al. pointed out that HPA axis function is reduced in patients with IBD [14]. There are also findings suggesting that a lower pituitary and adrenocortical activity are found in patients with functional gastrointestinal disorders [19]. These assumptions are notably relevant to stress induced increases in disease activity, although some studies undermined the primary role of dysregulations in the HPA in modulating IBS severity [20].

The role for the BGA in modulating neurodevelopment and behavior is highly supported by neuroimaging in patients with GI diseases in comparison to healthy population. Conducted studies highlighted among others increased activation of anterior insula, posterior insula and prefrontal cortex [21]. However, there are very few clinical neuroimaging studies on the effects of IBD and FGIDs on brain structure and function [16]. Further studies are needed to demonstrate the changes of specific brain structures in the course of these gastrointestinal disorders and our goal was to evaluate volume of subcortical structures.

## 2. Materials and Methods

All patients gave written consent to participate in the study. Study has been approved by The Bioethical Committee of The Military Medical Council (Street Koszykowa 78, 00-909 Warsaw) (document 107/12 dated 22 June 2012).

Patients suffering from FGIDs have been enrolled according to Rome IV Criteria. Patients with IBS and IBDs have been qualified according to the anamnesis and results of additional tests (colonoscopy with histopathological assessment, gastrofiberoscopy, capsule endoscopy and/or magnetic resonance enterography). Minimum period of three years from diagnosis has been established.

The exclusion criteria comprised lack of fulfillment of Rome IV Criteria for FGIDs, head trauma in anamnesis, severe additional diseases, depression, mental disorders, pregnancy and/or lactation and contraindications to MRI.

The study group included 18 patients with functional dyspepsia (FD), 20 with irritable bowel syndrome (IBS) and 19 with non-specific inflammatory bowel diseases (IBDs; Crohn’s disease and ulcerative colitis). The control group consisted of 19 healthy volunteers. See Table 1.

### 2.1. Scanning Protocol

The anatomical data sets were acquired in a 3T Achieva TX Scanner (Philips Healthcare, Best, The Netherlands) with the use of an 8-channel head coil. Examination protocol included standard T1 and T2 sequences to evaluate brain morphology and to exclude subjects with brain pathology which were further followed by 3D high-resolution T1 sequence (T1-TFE: TR = 7.44 ms TE = 3.6 ms, slice thickness: 1 mm, matrix 260 × 240, FOV = 260 × 240 (mm × mm). No contrast agent was administered.

### 2.2. Segmentation and Statistical Analyses

T1-weighted images were converted to nii format by MRIConvert (https://lcni.uoregon.edu/downloads/mriconvert (accessed on 5 September 2022)). Volumes of brain structures and cortical thickness were measured by freely available software FreeSurfer, version 6.0 [22] (Figure 1). FreeSurfer processing stream recon-all was used with 3t flag. Data were visually inspected. The volumes obtained from analyses were normalized to estimated total intracranial volume. The segmented volumes of each patient were visually checked by the medical physicist (AS) with 10 years of experience in this type of analysis in the field of neuroradiology.

The statistical analyses included the Shapiro–Wilk test to determine whether data was normally distributed. The majority of the analyzed volumes had normal data distribution thus, the comparisons between groups were performed with the one-way ANOVA and post-hoc tests with Bonfferoni correction. When the values was not normally distributed or when the parameter hadn’t passed the equality of variances Levene’s test then we used the Kruskall–Wallis test. *p*-values < 0.05 were considered significant. All the statistical analysis was carried out with SPSS Software ver. 26 (IBM, Chicago, IL, USA) [23].

## 3. Results

Table 2 presents the intergroup statistically significant differences between the IBD group and Control in the volume of the left thalamus, IBD group and IBS group in the volume of the left thalamus and IBD group vs Control group in the volume of the right thalamus. The IBD group shows the smallest thalamus volume of all groups (Figure 2). Detailed data are in Appendix A.

## 4. Discussion

There is very limited data about how gastrointestinal diseases affect the brain size. However, functional brain imaging has great promise in aiding our understanding of gastrointestinal pain neurophysiology and in the creation of models to investigate the effects of psychological factors and inflammation [24].

In this study, we found that there is a difference between the volume of the thalamus in IBD patients in comparison to both control group and IBS patients and it occurred to be smaller.

Cases of a decrease in gray matter volumes in dorsolateral prefrontal cortex and anterior midcingulate cortex (aMCC) were observed by Agostini et al. [25]. They also indicated that disease duration was negatively correlated with volumes of subgenual anterior cingulate (sACC), posterior MCC (pMCC), ventral posterior cingulate (vPCC), and parahippocampal cortices [25]. However, Bao et al. indicated that the gray matter (GM) volume in the CD patients were significantly higher in such structures as the putamen, pallidum, thalamus, hippocampal cortex, amygdala, precuneus, posterior parietal cortex, periaqueductal grey, and cerebellum, and at the same time were lower in many other cortical regions. In the same group of patients, the cortical thicknesses of the insula, cingulate cortex, parahippocampal cortex, and other cortical regions were also significantly reduced. What is more, the disease duration negatively correlated with the GM volumes of the right anterior cingulate cortex, dorsomedial prefrontal cortex, and left insula and the cortical thickness of the left insula and orbitofrontal cortex [26]. The above-mentioned results as well as other research [27,28,29,30] showing differences in activity of subcortical structures in inducted digestive tract pain, evaluating Default Mode Networks (DMNs) and the paper by Agostini et al. [25] on differences in grey matter volumes in patients with non-specific inflammatory bowel disease point to the disturbances of the brain-gut axis regulation.

However, our study confirmed only the association between the conduct of diseases such as IBD and IBS with the size of the thalamus, which in both cases has reduced volume. The human thalamus is described as a nuclear complex and relay center between the cerebral cortex and several subcortical brain regions located in the diencephalon. It supports both sensory and motor mechanisms [31]. Similar to our findings both Davis et al. [32] and Nair et al. [33] reported a reduction in thalamic volume in either IBS or IBD patients compared to controls. The important fact in the context of our study is that the thalamus is repeatedly associated with the dysfunction of brain-gut interaction [34]. There is also research on the role of disturbance of the brain-gut-axis in etiology of functional gastrointestinal disorders, referring to communication between the gut and the central nervous system [35,36].

Structural and functional alterations are seen as targets for evaluating or forecasting the effectiveness of treatment interventions meant to alleviate coexisting emotional and cognitive problems [37]. Many authors support the theory that stress, and other psychological disorders are factors in the development of diseases of the gastrointestinal tract as patients present with increased levels of anxiety, signs of depression and emotional discomfort [38,39]. We believe that a good direction for future research would be to include variables such as cognitive and emotional functioning and symptoms of depression. This will allow for a more holistic treatment of patients, thus ensuring a better quality of life—both in terms of physical and mental health.

## 5. Conclusions

This study indicated the association between the conduct of gastrointestinal diseases such as IBD and IBS with the size of the thalamus, which in both cases has reduced volume. The thalamus is repeatedly associated with the dysfunction of brain-gut interaction. Structural as well as functional changes are seen as targets for evaluating or forecasting the effectiveness of treatment interventions meant to alleviate coexisting emotional and cognitive problems.

## Figures and Tables

**Figure 1 diagnostics-12-02199-f001:**
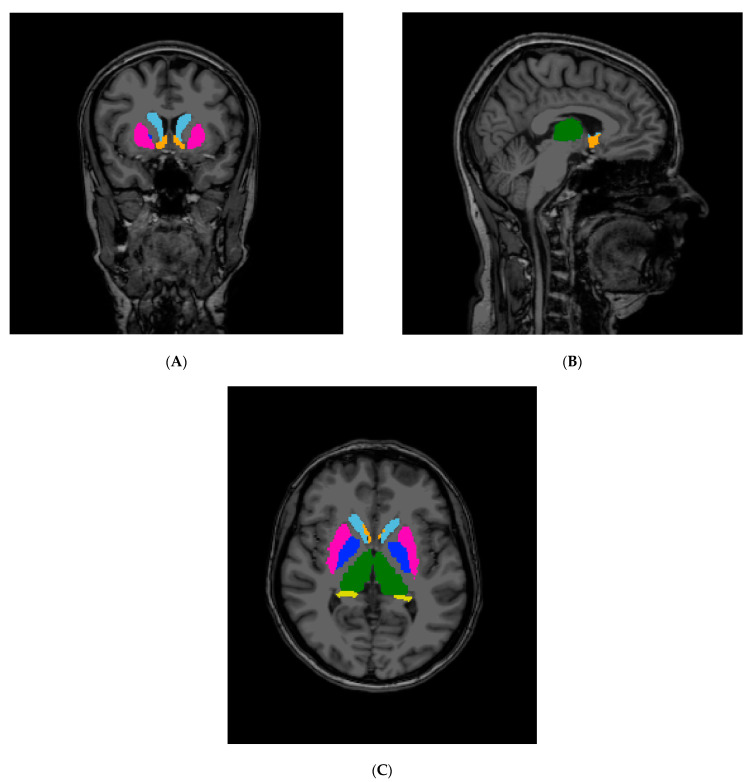
Brain structures reconstruction, FreeSurfer software. (**A**) coronal plane, (**B**) saggital plane, (**C**) axial plane; brain structures: pink—putamen, blue—pallidum, green—thalamus, yellow—hippocampus, orange—accumbens, light blue—caudate, turquoise—amygdala.

**Figure 2 diagnostics-12-02199-f002:**
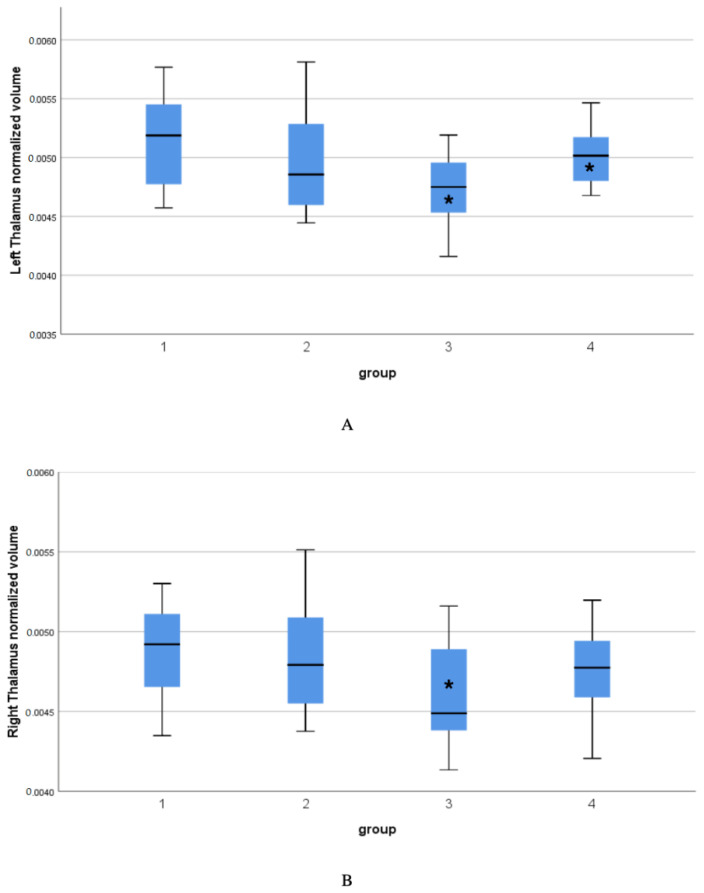
(**A**) the volume of the left thalamus in all groups, (**B**) the volume of the right thalamus in all group. Group: 1—control (CON), 2—functional dyspepsia (FD), 3—non-specific inflammatory bowel diseases (IBD), 4—irritable bowel syndrome (IBS). Error bars are indicating box plot. Asterisks are indicating significant differences between the IBD group and Control in the volume of the left thalamus, IBD group and IBS group in the volume of the left thalamus and IBD group vs Control group in the volume of the right thalamus.

**Table 1 diagnostics-12-02199-t001:** General characteristics of the study and control group. Group: 1—control (CON), 2—functional dyspepsia (FD), 3—non-specific inflammatory bowel diseases (IBD), 4—irritable bowel syndrome (IBS).

	Group 1 (CON)	Group 2 (FD)	Group 3 (IBD)	Group 4 (IBS)
N (71)	19	18	19	20
Age [years]	mean 34.15 (min. 24, max. 47) SD 8.01	mean 25.73 (min. 20, max. 40) SD 5.21	mean 31.73 (min. 21, max. 43) SD 5.82	mean 35.5 (min. 17, max. 62) SD 10.3
Sex (F=, M=)	F = 9, M = 10	F = 13, M = 5	F = 9, M = 10	F = 14, M = 6

**Table 2 diagnostics-12-02199-t002:** Brain structures volume analysis and comparisons between groups. Statistically significant differences between the IBD group and Control in the volume of the left thalamus, IBD group and IBS group in the volume of the left thalamus and IBD group vs Control group in the volume of the right thalamus are highlighted. (eTIV—estimated total intracranial volume, Levene’s test—test of homogeneity of variance, ANOVA—analysis of variance, CSF—cerebrospinal fluid). Detailed data are in Appendix A.

Volumes Normalized to eTIV	Levene’a *p*	ANOVA F	ANOVA p	Post-Hoc Contrast	Comments
Left cerebellum white matter	>>0.1	0.323	0.809	-	
Left cerebellum cortex	>>0.1	1.106	0.352	-	
Left thalamus	0.006—Kruskall–Wallis test performed	KW 10.611	KW *p* = 0.014	3->4 *p* = 0.015 3->1 *p* = 0.002	Non-parametric test
Left caudate	>>0.1	1.037	0.366	-	
Left putamen	>>0.1	0.849	0.472	-	
Left pallidum	>>0.1	2.346	0.08	-	
Left Hippocamp	>>0.1	0.421	0.738	-	
Left Amygdala	>>0.1	2.169	0.099	-	
Left Accumbens	>>0.1	0.664	0.577	-	
CSF	>>0.1	0.504	0.681	-	
Right cerebellum white matter	>>0.1	0.346	0.792	-	
Right cerebellum cortex	>>0.1	0.346	0.792	-	
Right thalamus	>>0.1	3.384	0.023	1->3 *p* = 0.022	
Right caudate	0.091	1.864	0.143	-	
Right putamen	>>0.1	0.887	0.452	-	
Right pallidum	>>0.1	1.648	0.186	-	
Right amygdala	>>0.1	1.381	0.256	-	
Right accumbens	>>0.1	0.858	0.467		
Right hippocamp	>>0.1	0.073	0.974	-	
CerebralWhiteMatter vol	>>0.1	0.687	0.563	-	
SubCortGrayVol	>>0.1	1.920	0.134	-	
TotalGrayVol	>>0.1	1.607	0.195	-	

## Data Availability

The data presented in this study are available on request from in the Appendix A.

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
