# Peer review of "Evaluation of Subcortical Structure Volumes in Patients with Non-Specific Digestive Diseases"

_diagnostics, 2022, doi:10.3390/diagnostics12092199_

Round 1

Reviewer 1 Report

This is a mono-center prospective study evaluating the subcortical volume in patients with non-specific digestive disease. The idea of ​​the study is interesting, but there are numerous shortcomings in the study's design, presentation, and interpretation of the results. The study itself was designed in 4 groups of subjects and one control group. The results in certain groups are not processed and interpreted in sufficient detail. The study's main result – the difference between IBD and control - is known from previous studies and does not bring anything new. However, a lot of changes are needed to improve the study.

Comments:

On the first page, at the footer, it is written Diagnostics 2021, and it should be 2022.

Introduction:

1) The introduction is too long, and the aim of the study is not clearly stated.

Materials and Methods:

1) There were no inclusion or exclusion criteria from the study

2) In Table 1, there are minor mistakes in writing the average of the years (somewhere there is a comma, and somewhere there is a period)

3) Also, the groups are not exactly equal in terms of sex because there are more than twice as many women in the FD and IBS group

4) It is not stated in the demographic data the duration of the disease and the BMI of the patients, which may affect the results

The results:

1) Table 2 is missing the results for the right Amygdala and right Accumbens

2) Table 2 is unclear because it is unclear where each group is, but only statistical data is given. The legend for statistical significance is not registered either

3) The text written below the table should be inside the text in the results and refer to figure 2.

4) In Figure 2, the legend does not explain which number represents a particular group

5) Figure 2 is not explained in the results text

Discussion:

1) It is not explained what could be the difference in volume between individual groups of respondents

2) It is not explained why there is a difference in the left and right thalamus between the IBD and the control group

3) The results were insufficiently analyzed and compared with other studies.

References:

References are double numbered

Author Response

Thank you for the review.

All changes has been uploaded to new manuscript and supplement as well in manuscript.

Introduction

Ad 1)  introduction has been shortened 

Materials and method:

Ad 1) inclusion and exclusion criteria - added to text

Patients suffering from FGIDs have been enrolled according to Rome IV Criteria. Patients with IBS and IBDs have been qualified according to the anamnesis and results of additional tests (colonoscopy with histopathological assessment, gastrofiberoscopy, capsule endoscopy and/or magnetic resonance enterography). Minimum period of three years from diagnosis has been established. 

The exclusion criteria comprised lack of fulfillment of Rome IV Criteria for FGIDs, head trauma in anamnesis, severe additional diseases, depression, mental disorders, pregnancy and/or lactation and contraindications to MRI. 

Ad 2) changed

Ad 3) that’s true - the groups are not equal (F and M), we had a problem to include equal groups - IBS and FD are more common in women than men

Ad 4) in inclusion criteria has been added that Minimum period of three years from diagnosis has been established. BMI - wasn’t counted.

Results:

Ad 1) added 

Ad 2) detailed information are given in extra file - as a „Supplement 1”  

Ad 3)  added to text in the results

Ad 4) added

Ad 5) added

Discussion and conslusion - changed 

References - done and new position added 

Reviewer 2 Report

This article explores the relationship between volumes of subcortical structures in patients with non-specific digestive diseases. This topic is interesting as the research about gut-brain axis is increasing and relationships between the gut microbiota and brain functionality is becoming more lucid. Though simple in its design, this study sheds some interesting thought as to how digestive disease can affect brain function by evaluating subcortical structure volumes. There are some major issues that need to be clarified before I can recommend this article for publication. In particular, the discussion and conclusion sections need to be rewritten to focus on the findings of this study. Please see below for more details.

1.)    There are some minor English language corrections that need to be addressed throughout the article. I will point out a few:

a.       I would rephrase the title: “Evaluation of subcortical structure volumes in patients with non-specific digestive diseases.”

b.       Introduction, 3rd paragraph: The diagnosis of IBS…

c.       Introduction, 3rd paragraph: Whereas functional dyspepsia is… = this sentence needs rewriting.

d.       Introduction, 5th paragraph: Not only it is pointed out that… = this sentence needs to be rewritten.

e.       Correction in 2.2 = T1 weighted images were converted to nifti format…

f.        Discussion 2nd to last paragraph: correction = grey matter density…not destiny.

2.)    The introduction covers a lot of background, which builds up the story of digestive disease and its effect on the brain. The last paragraph needs to be more clear stating what exactly you did in this study and what was the goal of this study instead of saying what needs to be done.

3.)    For the segmentation analysis, who did the inspection? Was it a trained radiologist? Was it more than one person? How did you make sure that the segmentations were all consistently evaluated between patients?

4.)    When you normalized the volumes to the intracranial volume, how was the intracranial volume estimated? Did that include the ventricular volume, or just the brain matter? How was the intracranial volume segmentation performed and how were the volume calculations made?

5.)    For all volume calculations, were the 2D segmentations rendered into a 3D volume or were the segmentation areas just added up to estimate a structure volume?

6.)    For the statistical analysis, why did you use one-way ANOVA? Typically, you would do that if you had an apriori knowledge of the expected effects. I feel in this case, it’s more appropriate to use a 2-way ANOVA since we don’t know beforehand whether the volumes of structures will be larger or smaller in disease compared to the healthy controls.

7.)    Table 2 title shouldn’t just mention the significant results. The title should inform what is presented in the whole table.

8.)    Both for Tables 1 and 2, all abbreviations must be explained in full form at the end of the table, for eg. eTIV. Even though they might be explained in the main text, the tables should stand by themselves with full form explanations of all abbreviations including the groups CON, FD, IBD, etc. in the footnotes on the table.

9.)    Table 1 should report the standard deviation of the ages for each group.

10.) Figure 2 should show asterisks indicating the significantly different groups. Also, the error bars are indicating standard deviation (SD) or standard error of the mean (SEM)? That should be mentioned in the figure legend and also the level of significance with p values. I would include the actual group names CON, FD, IBD, IBS just as a personal preference instead of having to back and forth and keeping track of what group number corresponds to each.

11.) I would add another figure showing a representative patient from the Con group, the IBD group, and the IBS group showing the thalamus sizes. Visual representations can be more convincing, but I leave that up to the authors as I am not sure how the unnormalized brain looks like in 2D segmentations. If there is a 3D rendered volume, that might be better. Just a thought.

12.) Claims are made in the discussion from this study and others that decrease in thalamus size could affect cognitive and other brain functions. Were the patients in this study assessed for depression, sleep abnormalities, or other psychological distress? That is something lacking here to complete the story and to make some constructive clinical conclusions about the thalamic volume effects. This would really strengthen the quality of this paper.

13.) The discussion section is quite lengthy explaining all sorts of connections between different brain substructures digestive diseases. In fact, there is no discussion of the results of this study except in the 2nd paragraph and the last sentence of the last paragraph. There is a lack of focus related to the findings of this study pertaining to the thalamus. This section needs to be rewritten to focus on the results of this study and how that relates to some of the findings already discussed.

14.) The conclusion is again vague and not directly connecting the findings of this study. That needs to be rewritten. Excess inflammation is mentioned here, for example, but there is no mention of imaging to visualize inflammation. So, the focus needs to be on the findings of this study first before suggesting other effects.

Author Response

Thank you for the review. 

Below changes added to manuscript, please see the attachment.

Ad 1)  A-F — done

Ad 2) added

Ad 3) The segmented volumes of each patient were visually checked by the medical physicist (A. S.) with 10 years of experience in this type of analysis in the field of neuroradiology.

  • added to text 

Ad 4 ) Freesurfer attempts to calculate intracranial volume from T1 data. It computes eTIV as all the structures inside the skull so ventricles are included. eTIV is a metrics which is automatically calculated in Freesurfer by dividing a predetermined constant with the factor by which the input T1w images are scaled in size to align to the template head. The calculation of eTIV originates from a method described by Buckner et al. [Buckner et al. (2004) NeuroImage 23:724-738]

Ad 5)  Instead of analyzing the brain as a 3D volume, however, FreeSurfer transforms the cortex into a 2D surface. The analysis in the Free surfer is prepared in two streams: surface-based stream and volume-based stream. Thus it is not so easy to say that areas are added up. The software construct models of the boundary between white matter and cortical gray matter as well as the pial surface. FreeSurfer uses the reconstructed surface, along with prior knowledge about the topology of a human brain, to label the cortical and subcortical structures. (https://surfer.nmr.mgh.harvard.edu/fswiki/FreeSurferAnalysisPipelineOverview) 

Ad 6) I hope that I understand your question properly. We used one-way ANOVA because we wanted to compare one independent variable that has multiple groups. In two-way ANOVA we should compare two factors. If we split the data into two factors (1st: healhy or disease and 2nd: all disease groups ) they want be independent. That's why one-way ANOVA better suits our hypothesis.

Ad 7)  done

Ad 8) done

Ad 9) done

Ad 10) Error bars are indicating box plot. Asterisks are indicating significant differences between the IBD group and Control in the volume of the left thalamus, IBD group and IBS group in the volume of the left thalamus and IBD group vs Control group in the volume of the right thalamus.

Ad 11) unfortunately we are not able to do this; there is no ability to create one figure from each group (‚the mean figure’) because values are counted from the values obtained from each person

Ad 12-14) Discussion and conclusion has been changed (also due to comments from second reviewer) and new references has been added as well.

Round 2

Reviewer 1 Report

The manuscript has been corrected according to the reviewer's comments and can be published in this form.

Reviewer 2 Report

Authors have addressed most of the issues I had raised.